# Exploration of Novel Scaffolds Targeting Cytochrome *b* of *Pyricularia oryzae*

**DOI:** 10.3390/ijms24032705

**Published:** 2023-01-31

**Authors:** Cecilia Pinna, Tommaso Laurenzi, Fabio Forlani, Luca Palazzolo, Claire Beatrice Nolan, Michael S. Christodoulou, Paolo Cortesi, Andrea Pinto, Ivano Eberini, Andrea Kunova, Sabrina Dallavalle

**Affiliations:** 1Department of Food Environmental and Nutritional Science (DeFENS), University of Milan, 20133 Milano, Italy; 2Department of Pharmacological and Biomolecular Sciences (DiSFeB), University of Milan, 20133 Milano, Italy; 3Data Science Research Center (DSRC), University of Milan, 20133 Milano, Italy

**Keywords:** high-throughput virtual screening, rice blast, antifungals, cytochrome *bc1* enzymatic inhibition, design and synthesis

## Abstract

The fulfilment of the European “Farm to Fork” strategy requires a drastic reduction in the use of “at risk” synthetic pesticides; this exposes vulnerable agricultural sectors—among which is the European risiculture—to the lack of efficient means for the management of devastating diseases, thus endangering food security. Therefore, novel scaffolds need to be identified for the synthesis of new and more environmentally friendly fungicides. In the present work, we employed our previously developed 3D model of *P. oryzae* cytochrome *bc1* (cyt *bc1*) complex to perform a high-throughput virtual screening of two commercially available compound libraries. Three chemotypes were selected, from which a small collection of differently substituted analogues was designed and synthesized. The compounds were tested as inhibitors of the cyt *bc1* enzyme function and the mycelium growth of both strobilurin-sensitive (WT) and -resistant (RES) *P. oryzae* strains. This pipeline has permitted the identification of thirteen compounds active against the RES cyt *bc1* and five compounds that inhibited the WT cyt *bc1* function while inhibiting the fungal mycelia only minimally. Serendipitously, among the studied compounds we identified a new chemotype that is able to efficiently inhibit the mycelium growth of WT and RES strains by ca. 60%, without inhibiting the cyt *bc1* enzymatic function, suggesting a different mechanism of action.

## 1. Introduction

*Pyricularia oryzae* Cavara is one of the most devastating pathogens worldwide [1]. Apart from being the major fungal pathogen of rice, causing rice blast, its host-specialized pathotypes also infect other cereal crops, such as wheat, causing a devastating wheat blast mainly in South America and South Asia [2,3,4], and turf grasses, causing gray leaf spot disease [5,6,7].

Agricultural production faces more and more restricted use of synthetic pesticides, which have already been drastically reduced in number over the past 20 years. However, to ensure a stable yield and to limit losses due to pests and pathogens, only a few efficient means with a low impact on the environment and human health are available. This is particularly true for the European risiculture sector, where after the withdrawal of tricyclazole from use in 2016 [8], only four fungicide classes are being used for the management of rice diseases and rice blast in particular. Strobilurins (quinone outside inhibitors, QoI, and FRAC 11) and demethylation inhibitors (DMI, FRAC 3) are synthetic fungicides that can be used in integrated disease management approaches, while sulphur (FRAC M02) and *Bacillus subtilis* QST 713 (FRAC BM02) can also be applied in biological agriculture [9].

However, such a limited portfolio of available products prompts the growers to repeatedly use the same compounds, especially on susceptible varieties largely grown in Europe, which in turn greatly increases the risk of resistance development in the pathogen populations. Fungicides with a single-site mode of action are especially at high risk of resistance development. This is also the case with QoI fungicides, which inhibit the electron transfer between cytochrome *b* and cytochrome *c* in the mitochondrial respiratory chain [10,11]. 

Strobilurins were introduced to the market in 1996 and still represent one of the most used fungicide classes [12,13,14]. Their mechanism of action has been studied in great detail; they act within the inner mitochondrial membrane, in particular on complex III by binding to the quinol oxidation site (Qo) of cytochrome *b*. This binding blocks the electron transfer between cytochrome *b* and cytochrome *c1*, which, in turn, leads to an energy deficiency in the fungal cells that halts oxidation of the reduced nicotinamide adenine dinucleotide (NADH) and synthesis of adenosine triphosphate (ATP), ultimately leading to cell death [15]. 

The resistance to strobilurins is most often determined by a single amino acid substitution from glycine to alanine (G143A), and until now, it has been detected in approximately 50 different pathogens [16,17]. Among these, strobilurin resistance in *P. oryzae* was first observed in strains isolated from perennial ryegrass, *Lolium perenne* [18], followed by its diffused presence in wheat isolates (*Triticum aestivum*) [2,19]. In rice isolates, QoI-resistance was first described in 2013 in Japan [20] and was absent from Europe until 2021, when it was detected in Italy [21]. This poses a threat to the European risiculture sector, as widespread distribution of such resistant strains would seriously compromise rice blast management with the available means.

The current situation urges the discovery and development of novel, highly effective compounds that would be able to control both wild-type (WT) and QoI-resistant (RES) populations of *P. oryzae*.

In our previous work, we developed and validated the first three-dimensional model of the *P. oryzae* cytochrome *bc1* (cyt *bc1*) complex containing azoxystrobin as a ligand [14]. Here, we exploit the model for the identification of compounds with novel scaffolds that would be able to inhibit mitochondrial respiration and consequently the mycelium growth in both WT and RES isolates of *P. oryzae*. 

## 2. Results

### 2.1. Virtual Screening and Identification of Potential Inhibitors of the Qo-Site of Cytochrome b

We employed the 3D model of the *P. oryzae* cyt *bc1* complex previously developed by our group [14] to perform a high-throughput virtual screening (HTVS) on two libraries of commercially available compounds, Agrochem and Biodesign, provided by Asinex (https://www.asinex.com accessed on 25 January 2023). The extra-precision glide score (XPG score) and its molecular weight-normalized value, as well as the binding mode and the number and type of molecular interactions, were used as selection criteria for the identification of the most promising structures. In particular, we narrowed our scope by focusing on molecules with the most favorable XPG score/mw and whose binding mode was characterized by the presence of relevant interactions resembling those of metyltetraprole [14]. This compound, a novel QoI fungicide with a unique tetrazolinone pharmacophore, was selected as a reference molecule as it is also active against QoI-resistant (RES) fungal pathogens [22,23]. From this procedure, eleven compounds emerged as potential hits for the development of antifungal agents (Appendix A). During this selection phase, the synthetic feasibility of the top-scoring structures was taken into consideration as well.

Eventually, based on the above considerations, we selected compounds **1a**, **2a**, and **3** (Figure 1) for their further implementation. Compound **3** is the result of a modification of the original library compound, where a fluorine atom was removed from the chlorobenzene ring.

### 2.2. Synthesis of Selected Compounds and Their Analogues

Compound **1a** displayed a strobilurin-like binding mode since its 4-fluorophenyl ring was placed near the Gly143 residue. Although its XPG score value was not among the most favourable, this molecule was associated with a relevant linearized XPG score/mw value, due to its low molecular weight. Interestingly, the compound showed a π-π interaction with Tyr132, similar to that formed by metyltetraprole [14], and an H-bond with the Tyr275 residue (Figure 1). Aiming at exploring the effect of different substitution patterns on the 6-phenylpyridazin3(*2H*)-one scaffold, compounds **1a**–**e** were synthesized simply by reacting the variously substituted acetophenones with glyoxylic acid and hydrazine in a two-step procedure (Figure 1) [24].

Compound **2a** was characterized by a strobilurin-like binding mode with an XPG score of −10.828 (Figure 1).

Three series of homologues (**2a**–**f**, **5a**–**d**, and **6a**–**h**, Figure 2) were designed and synthesized in order to investigate the role of the amide group and its substitution with an ester group, as well as the effects of the aliphatic chain length. Compounds **2a, 2d**, and **2e** were obtained via amide coupling by reacting the appropriate aniline with adipic or suberic acid, in the presence of HATU and DIPEA. The efforts to obtain **2b, 2c**, and **2f** by similar synthetic procedures involving the use of coupling reagents (HATU and EDCI) were not successful. These compounds were obtained by producing in situ adipoyl or suberoyl chloride, which was then reacted with 4-fluoroaniline or 2,4-dichloroaniline in the presence of triethylamine. 4-Halobenzoyl chlorides and aliphatic diamines (butanediamine and heptanediamine) afforded compounds **5a**–**d**, while esters **6a**–**h** were obtained by reacting the proper benzoyl chloride with propane-, butane-, hexane- and decanediol.

Hydrazone **3** displayed π-stacking interaction between the benzyl ring and Tyr280, while the carbonyl group interacted via H-bond with Glu273 (Figure 1). This promising pattern of interactions prompted us to synthesize **3** through the synthetic pathway shown in Figure 3. Benzylamine **7** was reacted with ethyl malonyl chloride **8** in the presence of TEA. Then, intermediate **9** was treated with hydrazine to afford the corresponding hydrazide (**10**), which was finally reacted with 2-chlorobenzaldehyde, giving a mixture of *cis*- and *trans*-isomers of compound **3.**

### 2.3. Enzymatic Inhibition of Cytochrome bc1 Complex by the Compounds

A total of 24 compounds were screened for the inhibitory action on the QoI-targeted cyt *bc1* mitochondrial complex (complex III). The cyt *bc1* function was evaluated in vitro by measuring the NADH-dependent cyt *c* reduction activity (NADH:cyt *c* oxidoreductase activity) of the mitochondrial fractions isolated from the strobilurin-sensitive A252 (WT), and -resistant PO21_01 (RES) *P. oryzae* strains. In this assay, the electron-donor quinol is the mitochondria-native ubiquinol, which is electron-recharged through the NADH-oxidation mediated by complex I.

As reported in Table 1 and Figure 2, the cyt *bc1* function of the WT strain mitochondrial fraction was clearly inhibited by five compounds (*p* < 0.01) and azoxystrobin. Interestingly, the cyt *bc1* activity of the RES mitochondrial fraction was inhibited by 13 compounds. As expected, azoxystrobin did not show (*p* > 0.01) inhibitory action on the cyt *bc1* function from the RES strain. The obtained results of the azoxystrobin activities on the mitochondrial fractions from the WT and RES strains validated the measured NADH:cyt *c* oxidoreductase activity as an indicator of the QoI-targeted cyt *bc1* function.

These results indicate compounds **3, 5a**, **6b**, and **6c** as a group of molecules of interest for their evident action on the cyt *bc1* function in both the WT and the RES strains (Figure 2). This subset of compounds was further tested for the inhibition of the cyt *bc1* function assayed specifically by measuring the decylubiquinol:cyt *c* oxidoreductase activity. This assay uses decylubiquinol (DBH_2_), a synthetic quinol, as the electron-donor substrate for the reduction of cyt *c* mediated by the mitochondrial cyt *bc1* (Table 2). Here, azoxystrobin (25 µM) also inhibited the DBH_2_:cyt *c* oxidoreductase activity of the mitochondrial fraction from the WT strain (*p* < 0.01), whereas no inhibition was evident in the RES strain. All the tested compounds (**3**, **5a, 6b**, and **6c**) showed inhibitory action (*p* < 0.01) on the DBH_2_:cyt *c* oxidoreductase activity of the mitochondrial fraction from the RES strain. Compounds **3** and **6c** inhibited the activity of the mitochondrial fraction from the WT strain (*p* < 0.05). 

### 2.4. Mycelium Growth Inhibition by the Compounds

The compounds **1a, 2a**, and **3** and their analogues were tested in parallel for their inhibitory activity against both QoI-sensitive (WT) and -resistant (RES) strains of *P. oryzae*. As the compounds were dissolved in DMSO, the effect of 1% DMSO on fungal growth was also assessed. It had minimal inhibitory activity on both the WT and the RES strains (<10% inhibition of mycelial growth, Appendix A). Thus, the inhibitory activity of DMSO 1% was subtracted from the inhibition calculation of the tested compounds (see Materials and Methods for details).

The overview of the inhibitory activity of all the tested compounds on the mycelial growth of the WT (A2.5.2.) and RES (PO21_01) strains of *P. oryzae* is summarized in Figure 3.

Azoxystrobin (AZX) showed >95% inhibition of the WT strain, while it showed only ca. 25% inhibition of the RES strain. Out of all tested compounds, only **1b** showed ca. 60% inhibition of both the WT and the RES strains of *P. oryzae*, and an additional six compounds (**1d**, **1e**, **2c**, **3**, **5a**, **b**) showed a weak activity (>10% inhibition) against both the WT and the RES strains.

In parallel, the compounds were screened on a panel of an additional six phytopathogenic fungi: *Penicillium expansum*, *Aspergillus niger*, *Mucor pyriformis*, *Botrytis cinerea*, *Fusarium culmorum*, and *F. graminearum* (Figure 4). In this case, the addition of 1% DMSO strongly inhibited the growth of *M. pyriformis* (ca. 40%), *B. cinerea* (ca. 20%), and *F. graminearum* (<15%), while it had only a minimum inhibitory effect on other fungi (<10%). Surprisingly, DMSO seemed to have a positive effect on the growth of *F. culmorum (*Appendix A).

Only compounds belonging to group **1** (Figure 4a), the analogues of **1a,** showed some activity against the tested fungi. None of the tested compounds inhibited *A. niger* and *B. cinerea*. Compound **1e** showed weak inhibitory activity (ca. 20% inhibition) against *P. expansum* and *M. pyriformis*. *Fusarium culmorum* was weakly inhibited by the compounds **1b** and **1d** (ca. 20% inhibition), while *F. graminearum* was the most inhibited pathogen. Its mycelium growth was reduced by compound **1a** by almost 40% and by its analogue **1b** by ca. 60%. Weak activity of **1d** and **1e** (almost 20% inhibition) against *F. graminearum* was also observed.

### 2.5. In Silico ADME/TOX Predictions

We performed a toxicokinetics (ADME/TOX) characterization of the compounds of interest, **1b**, **3, 5a**, **6b,** and **6c**, which showed activity on the cyt *bc1* function. Descriptors were calculated using the ACD/Percepta program suite. The ADME/TOX characterization is reported in Table 3: most of the tested compounds, except for **1b** and **6b**, showed poor oral bioavailability (%F), especially when taking into consideration the liver and gut first-pass effect. The apparent volume of distribution (Vd) values were determined together with the predicted tendency of each evaluated molecule to bind specific plasma proteins. The Vd values are well below the determined limit associated with a total body fat distribution (Vd > 200), suggesting a reduced level of lipophilicity and bioaccumulation. Considering the analyses involving the CYP450 superfamily, none of the selected molecules was predicted to undergo significant biotransformations due to the CYP450 isoforms considered by the software. However, the **6b** and **6c** diesters showed a potential inhibitory activity affecting the CYP450 isoform CYP1A2.

The toxicological endpoints considered of relevance for in silico hazard identification are reported in Table 4. All the molecules were shown to have the physicochemical properties suitable for blood–brain barrier penetration and potential CNS activity. An endocrine disruption risk assessment was evaluated based on the predicted binding affinity to estrogen receptor-α (ERα), and compounds **5a**, **6b**, and **6c** were predicted as weak ERα binders. None of the tested molecules was predicted to be an eye/skin irritant. Finally, aquatic life toxicity was assessed, an extremely relevant parameter for compounds intended to be released into the environment, such as pesticides and agrochemical active ingredients. 

## 3. Discussion

In this study, we employed a three-dimensional model of *P. oryzae* cytochrome *bc1* complex to screen two libraries of commercially available compounds (Agrochem and Biodesign). Virtual screening procedures implement rapid conformational search and empirical scoring functions to predict ligand–receptor interactions and the metrics proportional to the binding free energy. Usually, the predicted ligand affinities are within one order of magnitude of the experimental values [25]. The molecular interactions from the docking calculations may also be useful in predicting the mode of action and activity of the docked molecules [14,26]. Compounds with the best glide score and whose binding mode was characterized by the presence of relevant interactions were selected as promising novel scaffolds for the synthesis of a small collection of molecules, including 6-phenylpyridazin3(2H)-ones, aryl-amides, aryl-esters, and hydrazones. The obtained compounds were tested both by enzymatic assays with the isolated mitochondrial fractions containing the target of strobilurin fungicides (cyt *bc1*; complex III) and by biological assays on a panel of seven phytopathogenic fungi, including wild-type and the QoI-resistant strains of *P. oryzae*. 

The enzymatic assay based on the measurements of NADH:cyt *c* oxidoreductase activity was used to screen the compounds for the inhibition of the cyt *bc1* function. A strong inhibition of the WT mitochondrial fraction was already observed at 25 µM of azoxystrobin, while only a limited inhibition was detected on the RES mitochondrial fraction even at 200 µM (25% inhibition, *p* = 0.06), thus validating the NADH:cyt *c* oxidoreductase activity assay as an indicator of the strobilurin-targeted cyt *bc1* function.

The cyt *bc1* function of the mitochondrial fraction from both the *P. oryzae* WT and the RES strains was inhibited by five compounds. On the other hand, the RES mitochondrial fraction was inhibited by an additional eight compounds. In particular, both mitochondrial fractions (WT and RES) were inhibited by ca. 50% by the compounds **3**, **5a**, and **6b**,**c** in the NADH:cyt *c* oxidoreductase activity assay (Table 1, Figure 2). Considering altogether the enzyme inhibition results, the structural criteria that drove the selection of the compounds helped the discovery of new chemotypes with different sizes and shapes that were preferentially active on the cyt *bc1* in the *P. oryzae* RES strain. In particular, the compounds of series **1** are characterized by a rigid structure, whereas the compounds of series **5** and **6** have higher flexibility, and the chains with different lengths are tolerated.

The compounds **5a** and **6b** did not show the inhibition of the DBH_2_:cyt *c* oxidoreductase activity, whereas their inhibition of the NADH:cyt *c* oxidoreductase activity was evident. These different results could be explained by the use of the two diverse quinols involved in the enzyme reaction, the synthetic DBH_2_, or the mitochondrial ubiquinol. We hypothesize that the tested compounds could compete with the two quinols differently for the Qo site. 

Surprisingly, compounds **3**, **5a**, and **6b**,**c** showed only a minimal inhibition of the fungal mycelia, both in the WT and the RES strain of *P. oryzae*. This could indicate poor bioavailability, most likely due to various mechanisms, such as the detoxification by the fungal cell, the low absorption, or the inability to penetrate the cell wall, which impedes the molecule binding to the cyt *bc1* target.

On the other hand, compound **1b** displayed the best mycelium inhibition activity in vitro (ca. 60% inhibition) both in the WT and the RES *P. oryzae.* Compound **1b** differs from **1a** only by a chloro substituent at the para position in place of the fluorine, suggesting that the presence of a chlorine atom strongly enhances the biological activity of the 6-phenylpyridazin3(*2H*)-one core. Interestingly, **1b** showed only a low inhibition of the putative molecular target in enzymatic assays and a similar mycelium inhibition of both strobilurin-sensitive and -resistant strains of *P. oryzae.* These results together suggest that the **1b** biological activity is due to a different mechanism of action which will be investigated further.

A preliminary in silico ADME/TOX analysis was carried out, showing that the most promising compounds have a very reduced propensity for bioaccumulation in adipose tissue (low predicted Vd), which is a relevant hazard factor for potential pesticides; however, compounds **6(b,c)** were predicted as potentially strong CYP1A2 inhibitors and weak ERα binders (along with **5a**), suggesting that further and thorough in vitro/in vivo toxicological characterization is advisable. Summing up, the HTVS campaign using the 3D model of *P. oryzae* cytochrome *bc1* complex led to the identification of novel chemotypes that were potentially active on the cyt *bc1* enzyme in the strobilurin-sensitive and -resistant *P. oryzae*. Future efforts will focus on improving the physicochemical properties of the most active compounds, along with the conducting of formulation studies aimed at increasing their bioavailability.

## 4. Materials and Methods

### 4.1. Molecular Modelling, Virtual Screening, and ADME/TOX Analysis

The 3D model of the *P. oryzae* cytochrome *bc1* complex was built upon the crystallographic structure of bovine cyt *bc1* co-crystalized with azoxystrobin (PDB ID: 1SQB), as described in [14]. High-throughput virtual screening (HTVS) was performed using Glide [25,26] on the chemoteques Agrochem (13,524 molecules) and Biodesign (170,269 molecules), provided by Asinex (https://www.asinex.com accessed on 25 January 2023). Screening molecules were prepared using LigPrep (LigPrep, Schrödinger, LLC, New York, NY, USA, 2021) to generate relevant tautomeric states and enantiomers for each chiral centre. The HTVS protocol consisted of three subsequent stages applying flexible-ligand docking algorithms with increasing complexity (namely: HTVS, standard precision, and extra precision) to the top-scoring compounds from each run. The virtual screening resulted in 308 molecules, with a docking score (XPG score) ranging between −10.060 and −12.907 kcal/mol. The compounds were then clustered according to their interaction fingerprints (SIFT) and compared to the binding mode of methyltetraprole and the known active strobilurins [14]. The compounds were ranked after normalizing the docking XPG score by their molecular weight (MW). Because large molecules usually have the highest affinity, this metric, also known as “ligand efficiency”, provides an approximation of the affinity of the compounds by atomic mass units, which may be helpful in scoring compounds of varying sizes and identifying potential active molecules that are also within small MW ranges [27,28].

The ADME/TOX descriptors for the most promising compounds were predicted in silico using the ACD/Percepta (Advanced Chemistry Development, Inc., Toronto, ON, Canada, Pharma Algorithms, Inc., Toronto, ON, Canada) software suite. All the predictors computed by this software have been developed with the use of large, validated databases, QSAR models, and the integration of the expert knowledge of organic chemistry and toxicology.

### 4.2. Chemistry

All the reagents and solvents were purchased from commercial suppliers and used without any further purification. The ^1^H NMR and ^13^C NMR spectra were recorded with a Bruker AV600 (^1^H, 600 MHz; ^13^C, 150 MHz) spectrometer. The chemical shifts (δ) are expressed in ppm, and the coupling constants (*J*) are in Hz. All the reactions requiring anhydrous conditions were performed under a positive nitrogen flow, and all glassware was oven dried. Isolation and purification of the compounds were performed by flash column chromatography on silica gel 60 (230–400 mesh). TLC analyses were performed using commercial silica gel 60 F254 aluminium sheets. All the final products showed a purity > 95% by NMR spectroscopy.

### 4.3. General Procedures for the Synthesis of Compounds

#### 4.3.1. General Procedure for the Synthesis of 6-Phenylpyridazin-3(2H)-ones (**1a**–**e**)

In a round-bottom flask, the appropriate acetophenone (1 eq) was dissolved in glacial acetic acid. To the obtained solution (0.5 M), glyoxylic acid (1.2 eq) was added, and the reaction mixture was refluxed overnight. Then, the reaction was cooled to room temperature and a solution of NH_4_OH (32%) was added until pH 8 was reached. The aqueous phase was extracted three times with dichloromethane. Then, hydrazine hydrate (10 eq) was added to the aqueous layer and the reaction mixture was heated to reflux for two hours. The resulting precipitate was filtered and washed with water. 

*6-(4-Fluorophenyl)pyridazin-3(2H)-one* (**1a**): white powder, 50% yield. ^1^H NMR (600 MHz, CD_3_OD): δ 8.04 (d, *J* = 9.8 Hz, 1H), 7.93 (m, 2H), 7.23 (m, 2H), 7.08 (d, *J* = 9.8 Hz, 1H); ^13^C NMR (150 MHz, CD_3_OD): δ 165.1 (d, *J* = 249.0 Hz), 163.2, 146.4, 133.4, 132.4, 130.9, 129.3 (d, *J =* 8.9 Hz, 2C), 116.7 (d, *J* = 22.0 Hz, 2C).

*6-(4-Chlorophenyl)pyridazin-3(2H)-one* (**1b**): white powder, 70% yield. ^1^H NMR (600 MHz, CD_3_OD): δ 8.05 (d, *J* = 9.9 Hz, 1H), 7.90 (d, *J* = 8.8 Hz, 2H), 7.56 (d, *J* = 8.8 Hz, 2H), 7.01 (d, *J* = 9.9 Hz, 1H); ^13^C NMR (150 MHz, DMSO-d_6_): δ 160.1, 124.7, 133.9, 133.5, 131.3, 130.2, 128.9 (2C), 127.4 (2C).

*6-(2,5-Dimethoxyphenyl)pyridazin-3(2H)-one* (**1c**): white powder, 60% yield. ^1^H NMR (600 MHz, CD_3_OD): δ 7.90 (d, *J* = 9.8 Hz, 1H), 7.14 (d, *J* = 3 Hz, 1H), 7.07 (d, *J* = 8.9 Hz, 1H), 7.01 (dd, *J* = 3.0, 8.9 Hz, 1H), 6.97 (d, *J* = 9.8, 1H), 3.84 (s, 3H), 3.80 (s, 3H); ^13^C NMR (150 MHz, CD_3_OD): δ 163.3, 155.3, 152.7, 147.2, 137.6, 128.8, 126.1, 117.2, 116.2, 114.0, 56.7, 56.2. 

*6-(4-(Trifluoromethyl)phenyl)pyridazin-3(2H)-one* (**1d**): light brown powder, 60% yield. ^1^H NMR (600 MHz, CD_3_OD): δ 8.08 (m, 3H), 7.78 (d, *J* = 8.2 Hz, 2H), 7.10 (d, *J* = 9.9 Hz, 1H); ^13^C NMR (150 MHz, CD_3_OD): δ 163.2, 145.7, 139.6, 133.2, 132.2 (q, *J* = 22.0 Hz), 131.0, 127.6 (2C), 126.8 (2C), 125.5 (q, *J* = 272.0 Hz).

*6-([1,1’-Biphenyl]-4-yl)pyridazin-3(2H)-one* (**1e**): light brown powder, 10% yield. ^1^H NMR (600 MHz, DMSO-*d*_6_): δ 8.10 (d, *J* = 9.8 Hz, 1H), 7.96 (d, *J* = 8.5 Hz, 2H), 7.79 (d, *J* = 8.5, 2H), 7.73 (d, *J* = 7.8 Hz, 2H), 7.49 (t, *J* = 7.8 Hz, 2H), 7.40 (t, *J* = 7.8 Hz, 1H), 7.01 (d, *J* = 9.8 Hz, 1H); ^13^C NMR (150 MHz, DMSO-*d*_6_): δ 160.2, 143.4, 140.7, 139.3, 133.6, 131.4, 130.1, 129.0 (2C), 127.8, 127.1 (2C), 126.6 (2C), 126.2 (2C).

#### 4.3.2. General Procedure for the Synthesis of Compounds **2b**, **2c**, and **2f**

Suberic or adipic acid (2 mmol) was dissolved in thionyl chloride (1.45 mL, 10 eq) under a nitrogen atmosphere in a round-bottom flask. Three drops of dimethylformamide were added, and the reaction mixture was heated to reflux for 90 min. After complete consumption of the starting material, the reaction mixture was cooled to room temperature, diluted with a 1:1 toluene/heptane mixture, and evaporated under reduced pressure. This procedure was repeated two more times until the excess of thionyl chloride was completely removed. The corresponding acyl chloride was obtained in quantitative yield, and it was used for the next step without further purification. The obtained adipoyl/suberoyl chloride was dissolved in anhydrous dichloromethane (15 mL) under a nitrogen atmosphere, and the solution was cooled to 0 °C in an ice bath. The appropriate anilines (2.2 eq) (4-fluoroaniline for **2b** and 2,4-dichloroaniline for **2c** and **2f**) and triethylamine (4 eq) were added; then, the reaction mixture was warmed to room temperature and stirred for 24 h. The resulting precipitate was filtered and washed with water, giving the desired products.

*N^1^,N^8^-bis(4-fluorophenyl)octanediamide* (**2b**): white solid, 70% yield. ^1^H NMR (600 MHz, DMSO-*d_6_*): δ 9.95 (s, 2H), 7.61 (m, 4H), 7.11 (m, 4H), 2.28 (t, *J* = 7.4 Hz, 4H), 1.59 (m, 4H), 1.32 (m, 4H). ^13^C NMR (150 MHz, DMSO-*d_6_*): δ 171.1 (2C), 157.8 (d, *J* = 249.0 Hz, 2C), 135.7 (2C), 120.7 (d, *J* = 9.0 Hz, 4C), 115.2 (d, *J* = 22.0 Hz, 4C), 36.3 (2C), 28.5 (2C), 25.0 (2C).

*N^1^,N^8^-bis(2,4-dichlorophenyl)octanediamide* (**2c**): white solid, 60% yield. ^1^H NMR (600 MHz, DMSO-*d*_6_): δ 9.52 (s, 2H), 7.70 (d, *J =* 8.8 Hz, 2H), 7.64 (d, *J* = 2.10 Hz, 2H), 7.39 (dd, *J* = 8.8 Hz, *J* = 2.10 Hz, 2H), 2.34 (t, *J* = 6.9 Hz, 4H), 1.60 (m, 4H), 1.36 (m, 4H). ^13^C NMR (150 MHz, DMSO-*d*_6_): δ 171.7 (2C), 134.2 (4C), 129.2 (2C), 128.8 (2C), 127.4 (4C), 35.6 (2C), 28.3 (2C), 25.0 (2C).

*N^1^,N^8^-bis(2,4-dichlorophenyl)adipamide* (**2f**): white solid, 50% yield. ^1^H NMR (600 MHz, DMSO-*d*_6_): δ 9.54 (s, 2H), 7.72 (d, *J* = 8.2, Hz, 2H), 7.65 (d, *J* = 2.1 Hz, 2H), 7.40 (dd, *J* = 8.2, 2.1 Hz, 2H), 2.42 (m, 4H), 1.65 (m, 4H). ^13^C NMR (150 MHz, DMSO-*d*_6_): δ 171.5 (2C), 134.2 (4C), 129.3 (2C), 128.8 (2C), 127.5 (4C), 35.5 (2C), 24.8 (2C).

#### 4.3.3. General Procedures for the Synthesis of Compounds **2a**, **2d**, and **2e**

In a round-bottom flask, suberic or adipic acid (1 mmol) was dissolved in 8 mL of anhydrous THF under a nitrogen atmosphere. 1-[Bis(dimethylamino)methylene]-1H-1,2,3-triazolo [4,5-b] pyridinium 3-oxide hexafluorophosphate (HATU, 2.2 eq) and diisopropylamine (2.2 eq) were added and the reaction mixture was stirred at room temperature for 30 min. Then, 2 equivalents of the selected aniline (4-chloro or 4-fluoroaniline) were added, and the obtained solution was stirred at room temperature for 16 h. After reaction completion, the obtained solid was filtered and washed with water. No further purification was needed.

*N^1^,N^8^-bis(4-chlorophenyl)octanediamide* (**2a**): white solid, 60% yield. ^1^H NMR (600 MHz, DMSO-*d*_6_): δ 9.98 (s, 2H), 7.62 (d, *J* = 8.7 Hz, 4H), 7.32 (d, *J* = 8.7 Hz, 4H), 2.29 (t, *J* = 7.4 Hz, 4H), 1.59 (m, 4H), 1.32 (m, 4H). ^13^C NMR (150 MHz, DMSO-*d*_6_): δ 171.4 (2C), 138.3 (2C), 128.5 (4C), 126.4 (2C), 120.5 (4C), 36.3 (2C), 28.4 (2C), 24.9 (2C).

*N^1^,N^8^-bis(4-chlorophenyl)adipamide* (**2d**): white solid, 50% yield. ^1^H NMR (600 MHz, DMSO-*d*_6_): δ 10.04 (s, 2H), 7.62 (d, *J* = 8.7 Hz, 4H), 7.33 (d, *J* = 8.7 Hz, 4H), 2.33 (m, 4H), 1.62 (m, 4H). ^13^C NMR (150 MHz, DMSO-*d*_6_): 171.3 (2C), 138.3 (2C), 128.5 (4C), 126.4 (2C), 120.5 (4C), 36.2 (2C), 24.7 (2C).

*N^1^,N^6^-bis(4-fluorophenyl)adipamide* (**2e**): white solid, 55% yield. ^1^H NMR (600 MHz, DMSO-*d*_6_): δ 9.88 (s, 2H), 7.55 (m, 4H), 7.06 (m, 4H), 2.27 (m, 4H), 1.58 (m, 4H). ^13^C NMR (150 MHz, DMSO-*d*_6_): *δ* 170.9 (2C), 157.8 (d, *J* = 237.0 Hz), 135.7 (2C), 120.8 (d, *J* = 8.9 Hz, 4C), 115.1 (d, *J* = 22.0 Hz, 4C), 36.1(2C), 24.9 (2C).

#### 4.3.4. General Procedure for the Synthesis of Compounds **5a**–**d**

A solution of butanediamiane or heptanediamine (1 mmol) in dry dichloromethane (2.5 mL) was prepared in a round-bottom flask under a nitrogen atmosphere, and then, triethylamine (4 eq) was added. In the meantime, a solution of 4-halobenzoyl chloride (2 mmol) in anhydrous dichloromethane (2.5 mL) was prepared and then added dropwise to the former solution. The reaction mixture was stirred at room temperature until the starting materials were completely consumed (2–3 h). The obtained white precipitate was filtered and washed with NaHCO_3_ (soln. 5%), HCl 0.5 M, and water.

*N,N’-(heptane-1,7-diyl)bis(4-chlorobenzamide)* (**5a**): white solid, 30% yield. ^1^H NMR (600 MHz, DMSO-*d*_6_): δ 8.50 (t, *J* = 5.5 Hz, 2H), 7.84 (d, *J* = 8.3 Hz, 4H), 7.52 (d, *J* = 8.3 Hz, 4H), 3.23 (m, 4H), 1.51 (m, 4H), 1.31 (m, 6H). ^13^C NMR (150 MHz, DMSO-*d*_6_): δ 165.0 (2C), 135.8 (2C), 133.4 (2C), 129.1 (4C), 128.3 (4C), 39.5 (2C overlapped to the solvent signal), 29.0 (2C), 28.5, 26.4 (2C).

*N,N’-(heptane-1,7-diyl)bis(4-fluorobenzamide)* (**5b**): white solid, 80% yield. ^1^H NMR (600 MHz, DMSO-*d*_6_): δ 8.45 (t, *J* = 5.2 Hz, 2H), 7.98 (m, 4H), 7.27 (m, 4H), 3.25 (m, 4H), 1.51 (m, 4H), 1.31 (m, 6H). ^13^C NMR (150 MHz, DMSO-*d*_6_): δ 165.0 (2C), 163.7 (d, *J* = 248.0 Hz, 2C), 131.2 (2C), 129.7 (d, *J* = 9.0 Hz, 4C), 115.1 (d, *J* = 22.0 Hz, 4C), 39.5 (2C overlapped to the solvent signal), 29.0 (2C), 28.5, 26.5 (2C).

*N,N’-(butane-1,4-diyl)bis(4-chlorobenzamide)* (**5c**): white solid, 90% yield. ^1^H NMR (600 MHz, DMSO-*d*_6_): δ 8.54 (t, *J* = 5.6 Hz, 2H), 7.85 (d, *J* = 8.4 Hz, 4H), 7.52 (d, *J* = 8.4 Hz, 4H), 3.28 (m, 4H), 1.56 (m, 4H). ^13^C NMR (150 MHz, DMSO-*d*_6_): δ 165.0, 135.8, 133.4, 129.1, 128.3, 39.0, 26.6.

*N,N’-(butane-1,4-diyl)bis(4-fluorobenzamide)* (**5d**): white solid, 65% yield. ^1^H NMR (600 MHz, DMSO-*d*_6_): δ 8.48 (t, *J* = 5.3 Hz, 2H), 7.90 (m, 4H), 7.28 (m, 4H), 3.28 (m, 4H), 1.56 (m, 4H). ^13^C NMR (150 MHz, DMSO-*d*_6_): δ 165.0 (2C), 163.7 (d, *J* = 248.1 Hz, 2C), 131.1 (2C), 129.7 (d, *J* = 8.8 Hz, 4C), 115.1 (d, *J* = 21.5 Hz, 4C), 39.5 (2C overlapped to the solvent signal), 26.7 (2C).

#### 4.3.5. General Procedure for the Synthesis of Compounds **6a**–**h**

In a round-bottom flask, the selected diol (1 mmol) was dissolved in anhydrous toluene (10 mL) under a nitrogen atmosphere. Triethylamine (4 eq) was added, followed by the proper 4-halobenzoyl chloride (2 eq). The reaction mixture was stirred under reflux for 2–16 h. After reaction completion, the solvent was evaporated under reduced pressure. The crude product was purified by flash chromatography (see below).

*Propane-1,3-diyl bis(4-chlorobenzoate)* (**6a**). The crude product was purified by flash chromatography (cyclohexane/ethyl acetate 9:1), giving the desired product as a white solid in 96% yield. ^1^H NMR (600 MHz, CDCl_3_): δ 7.94 (d, *J* = 8.5 Hz, 4H), 7.38 (d, *J* = 8.5 Hz, 4H), 4.49 (t, *J* = 6.2 Hz, 4H), 2.25 (m, 2H). ^13^C NMR (150 MHz, CDCl_3_): δ 165.7 (2C), 139.6 (2C), 131.1 (4C), 128.8 (4C), 128.6(2C), 62.1 (2C), 28.3.

*Propane-1,3-diyl bis(4-fluorobenzoate)* (**6b**). The crude product was purified by flash chromatography (cyclohexane/ethyl acetate 9:1), giving the desired product as a white solid in 97% yield. ^1^H NMR (600 MHz, CDCl_3_): δ 8.04 (m, 4H), 7.09 (m, 4H), 4.49 (t, *J* = 6.2 Hz, 4H), 2.25 (m, 2H). ^13^C NMR (150 MHz, CDCl_3_): δ 165.8 (d, *J* = 254.0 Hz, 2C), 165.5 (2C), 132.1 (d, *J* = 9.7 Hz, 4C), 126.3 (2C), 115.5 (d, *J* = 22.4 Hz, 4C), 61.8 (2C), 28.2.

*Butane-1,4-diyl bis(4-chlorobenzoate)* (**6c**). The crude product was purified by flash chromatography (cyclohexane/ethyl acetate 85:15 → 50:50), giving the desired product as a white solid in 55% yield. ^1^H NMR (600 MHz, CDCl_3_): δ 7.96 (d, *J* = 8.5 Hz, 4H), 7.41 (d, *J* = 8.5 Hz, 4H), 4.39 (m, 4H), 1.94 (m, 4H). ^13^C NMR (150 MHz, CDCl_3_): δ 165.8 (2C), 139.6 (2C), 131.1 (4C), 128.9 (4C), 64.8 (2C), 25.6 (2C).

*Butane-1,4-diyl bis(4-florobenzoate)* (**6d**). The crude product was purified by flash chromatography (cyclohexane/ethyl acetate 85:15 → 50:50), giving the desired product as a white solid in 55% yield. ^1^H NMR (600 MHz, CDCl_3_): δ 8.05 (m, 4H), 7.10 (m, 4H), 4.39 (m, 4H), 1.94 (m, 4H). ^13^C NMR (150 MHz, CDCl_3_): δ 165.74 (d, *J* = 253.0 Hz, 2C), 165.72 (2C), 132.2 (d, *J* = 9.3 Hz, 4C), 126.6 (2C), 115.6 (d, *J* = 22.0 Hz, 4C), 64.7 (2C), 25.7 (2C).

*Hexane-1,6-diyl bis(4-chlorobenzoate)* (**6e**). The crude product was purified by flash chromatography (cyclohexane/ethyl acetate 95:5), giving the desired product as a white solid in 55% yield. ^1^H NMR (600 MHz, CDCl_3_): δ 7.96 (d, *J* = 8.6 Hz, 4H), 7.40 (d, *J* = 8.6 Hz, 4H), 4.32 (t, *J* = 6.7 Hz, 4H), 1.80 (m, 4H), 1.46 (m, 4H). ^13^C NMR (150 MHz, CDCl_3_): δ 165.7 (2C), 139.2 (2C), 130.8 (4C), 128.8 (2C), 128.6 (4C), 65.0 (2C), 28.5 (2C), 25.7 (2C).

*Hexane-1,6-diyl bis(4-florobenzoate)* (**6f**). The crude product was purified by flash chromatography (cyclohexane/ethyl acetate 95:5), giving the desired product as a white solid in 65% yield. ^1^H NMR (600 MHz, CDCl_3_): δ 8.04 (m, 4H), 7.10 (m, 4H), 4.32 (t, *J* = 6.3 Hz, 4H), 1.80 (m, 4H), 1.53 (m, 4H). ^13^C NMR (150 MHz, CDCl_3_): 165.66 (d, *J* = 254.0 Hz, 2C), 165.62 (2C), 132.0 (d, *J* = 9.3 Hz, 4C), 126.6 (2C), 115.4 (d, *J* = 22.0 Hz, 4C), 65.0 (2C), 28.6 (2C), 25.7 (2C).

*Decane-1,10-diyl bis(4-chlorobenzoate*) (**6g**). The crude product was purified by flash chromatography (cyclohexane/ethyl acetate 95:5), giving the desired product as a white solid in 40% yield. ^1^H NMR (600 MHz, CDCl_3_): δ 7.97 (d, *J* = 8.5 Hz, 4H), 7.40 (d, *J* = 8.5 Hz, 4H), 4.30 (t, *J* = 6.8 Hz, 4H), 1.75 (m, 4H), 1.42 (m, 4H), 1.34 (m, 8H). ^13^C NMR (150 MHz, CDCl_3_): 165.7 (2C), 139.1 (2C), 130.8 (4C), 128.9 (2C), 128.6 (4C), 65.3 (2C), 29.3 (2C), 29.1 (2C), 28.6 (2C), 25.9 (2C).

*Decane-1,10-diyl bis(4-florobenzoate*) (**6h**). The crude product was purified by flash chromatography (cyclohexane/ethyl acetate 9:1) and then recrystallized from hexane. **4f** was obtained as a white solid in 70% yield. ^1^H NMR (600 MHz, CDCl_3_): δ 8.05 (m, 4H), 7.10 (m, 4H), 4.30 (t, *J* = 6.8 Hz, 4H), 1.76 (m, 4H), 1.43 (m, 4H), 1.34 (m, 8H). ^13^C NMR (150 MHz, CDCl_3_): δ 165.9 (2C), 165.8 (d, *J* = 252.0 Hz, 2C), 132.2 (d, *J* = 9.3 Hz, 4C), 126.9 (2C), 115.6 (d, *J* = 23.0 Hz, 4C), 65.4 (2C), 29.6 (2C), 29.4 (2C), 28.9 (2C), 26.2 (2C).

#### 4.3.6. Synthesis of Ethyl 3-(Benzylamino)-3-oxopropanoate (**9**)

Ethyl malonyl chloride (2 mmol) was dissolved in dry dichloromethane (5 mL) in a round-bottom flask under a nitrogen atmosphere. Benzylamine (1 eq) and triethylamine (1.5 eq) were added to the solution, and the reaction mixture was stirred at room temperature for 3 h. After completion, the reaction mixture was washed with HCl 1M and water. The organic layer was dried over anhydrous Na_2_SO_4_, and the solvent was evaporated. The crude product was purified by flash chromatography (cyclohexane/ethyl acetate 8:2 → 7:3), giving **9** as a colorless oil in 55% yield. ^1^H NMR (600 MHz, CDCl_3_): δ 7.45 (bs, 1H), 7.34 (m, 2H), 7.29 (m, 3H), 4.47 (d, *J* = 5.6 Hz, 2H), 4.19 (q, *J* = 7.1 Hz, 2H), 3.36 (s, 2H), 1.28 (t, *J* = 7.1 Hz, 3H).

#### 4.3.7. N-Benzyl-3-hydrazinyl-3-oxopropanamide (**10**)

Compound **7** (0.904 mmol) was dissolved in ethanol (2 mL) and treated with hydrazine hydrate (2 eq). The reaction mixture was heated to reflux and stirred until the starting material was consumed (16 h). Then, the reaction mixture was cooled to room temperature, and the obtained white precipitate was filtered (167 mg, 89% yield) and used for the next step without further purification. ^1^H NMR (600 MHz, DMSO-*d*_6_): δ 9.12 (s, 1H), 8.46 (m, 1H), 7.29 (m, 5H), 4.29 (d, *J* = 6.0 Hz, 2H), 4.26 (bs, 2H), 3.03 (s, 2H).

#### 4.3.8. N-Benzyl-3-(2-(2-chlorobenzylidene) Hydrazinyl)-3-oxopropanamide (**3**)

2-chloro benzaldehyde (1 eq) was added to a suspension of N-benzyl-3-hydrazinyl-3-oxopropanamide **9** (0.240 mmol) in ethanol (5 mL), and the reaction mixture was heated to reflux for 6 h. Then, the solvent was evaporated. The crude product was purified by flash chromatography (dichloromethane/methanol 98:2). (*E*)-N-benzyl-3-(2-(2-chlorobenzylidene) hydrazinyl)-3-oxopropanamide and (*Z*)-N-benzyl-3-(2-(2-chlorobenzylidene) hydrazinyl)-3-oxopropanamide were obtained, as a mixture, in 51% and 25% yields, respectively. The following ^1^H NMR and ^13^C NMR are referred to the 1:2 mixture of (*E*)-N-benzyl-3-(2-(2-chlorobenzylidene) hydrazinyl)-3-oxopropanamide and (*Z*)-N-benzyl-3-(2-(2-chlorobenzylidene) hydrazinyl)-3-oxopropanamide:

^1^H NMR (600 MHz, DMSO-*d*_6_): *δ* 11.8 (bs, 1H, minor isomer), 11.6 (bs, 1H, major isomer), 8.59 (t, *J* = 5.6 Hz, 1H, minor isomer), 8.58 (s, 1H, minor isomer), 8.57 (t, *J* = 5.9 Hz, 1H, major isomer), 8.36 (s, 1H, major isomer), 7.97–7.93 (m, 1H minor isomer + 1H major isomer), 7.55–7.50 (m, 1H minor isomer + 1H major isomer), 7.45 (td, *J* = 1.8, 7.5, Hz, 1H, minor isomer), 7.44–7.40 (m, 2H, major isomer), 7.35–7.28 (m, 2H major isomer + 2H minor isomer), 7.27–7.24 (m, 2H minor isomer + 1H major isomer), 7.23–7.18 (m, 2H minor isomer + 2H major isomer), 4.31 (d, *J* = 6.2 Hz, 2H, minor isomer), 4.29 (d, 6.1 Hz, 2H, major isomer), 3.60 (s, 2H, major isomer), 3.32 (s, 2H, minor isomer).

^13^C NMR (150 MHz, DMSO-*d*_6_): *δ* 169.4, 166.4, 166.1, 163.4, 142.5, 139.2 (2C), 138.8, 133.1, 132.9, 131.4 (3C), 131.1, 129.8 (2C), 128.3 (2C), 128.1 (2C), 127.6 (2C), 127.5 (4C), 127.2 (4C), 42.9, 42.2 (2C), 41.6.

### 4.4. Fungal Strains

In this study, strains belonging to seven different species were used. *Pyricularia oryzae* A2.5.2 (sensitive to strobilurins; WT) [29] and PO21_01 (resistant to strobilurins, RES), *Penicillium expansum* EM22, *Mucor pyriformis* EM14, *Botrytis cinerea* EM1, *Fusarium culmorum* FcUK [30], and *F. graminearum* ITA1601 [31] made a part of a fungal collection and were maintained in the laboratory of plant pathology of the University of Milan; *Aspergillus niger* ITEM9076 and ITEM9077 were obtained from the Agro-Food Microbial Culture Collection ITEM (Institute of Sciences of Food Production—ISPA, CNR, Bari, Italy). The strains were maintained as single-spore isolates on a malt-agar medium (MA: 20 g/L malt extract, Oxoid, UK; 15 g/L agar, Oxoid, UK) at 4 °C.

### 4.5. Enzyme Inhibition Assays for the Measurement of QoI Action

*P. oryzae* mitochondrial fractions from the A2.5.2 and PO21_01 strains were prepared as described previously [11], with the following modification. The final collected mitochondrial pellet was suspended in mitochondrial buffer (3 mL; 0.5 M sucrose, 10 mM KH_2_PO_4_, 10 mM KCl, 10 mM MgCl_2_, 0.2 mM EDTA, pH 7.2), divided into 3 Eppendorf tubes, and centrifuged for 20 min at 4 °C, at 10000 rpm (rotor F45–24-11, Eppendorf). Each pellet was resuspended in 0.2 mL of the same buffer and stored at −80 °C until use.

The strobilurin-like (QoI) action was evaluated by measuring the cyt *bc1*-mediated cyt *c* reduction in the presence of the tested compounds (200 µM) and the isolated mitochondrial fractions. The cyt *c* reduction activity was measured through a spectrophotometrical assay (λ = 550 nm) based on the synthetic quinol, decylubiquinol (DBH_2_), for the determination of the DBH_2_:cyt *c* oxidoreductase activity described in [11].

The cyt *bc1*-mediated cyt *c* reduction was also detected by measuring the NADH:cyt *c* oxidoreductase activity. The measured activity is based on the native mitochondrial ubiquinol as the quinol substrate whose electron recharging is ensured by the coupled NADH-oxidation reaction of the mitochondrial complex I. The assay was carried out as described for the DBH_2_:cyt *c* oxidoreductase activity assay, replacing DBH_2_ with NADH (25 μL of 4 mM stock, freshly prepared in 0.05 M Tris-HCl, 1 mM EDTA, pH 7.7). Azoxystrobin (31697, Sigma Aldrich, Milano, Italy) was used as the reference QoI. In both assays, one enzyme unit (U) is the amount that yields one micromole of reduced cyt *c* under the assay conditions (ε^550^ = 16.3 mM^−1^ cm^−1^), and for the inhibition studies, the enzyme rate in the presence of the tested molecule (rate_molecule_) was compared to that achieved in the presence of the compound-diluent DMSO (rate_DMSO_), in order to calculate the percent inhibition (I%) as follows:I (%) = (rate_DMSO_ − rate_molecule_/rate_DMSO_) × 100(1)

### 4.6. In Vitro Fungal Mycelium Growth Inhibition by the Compounds

The inhibitory activity of the novel compound fungicides on the mycelium growth of the different fungi was evaluated as described previously [11,32], with slight modifications. The fungi were divided into two groups: fast- and medium/slow-growing. The fast-growing fungi were *B. cinerea* EM1, *M. pyriformis* EM14, *F. culmorum* FcUK, and *F. graminearum* ITA1601. The medium/slow-growing fungi included *P. oryzae* A2.5.5 and PO21_01, *P. expansum* EM22, and *A. niger* ITEM9076 and ITEM9077. A mycelium plug (0.5 cm in diameter) obtained from an actively growing fungal colony was transferred to MA medium plates supplemented or not with tested compounds at the concentration of 25 mg/L in two biological replicates. Due to the low solubility of the tested dual molecules in water, they were dissolved in dimethyl sulfoxide (DMSO). Therefore, two controls were included: the MA medium (C, control for azoxystrobin) and the MA medium supplemented with DMSO at the final concentration of 1% *v*/*v* (C, control for tested compounds). The plates were incubated at 24 °C in the dark. The mycelium growth was measured for fast-growing fungi at 2–4 days after inoculation (dai), and for the medium/slow-growing fungi at 6 and 7 dai. The inhibition of mycelium growth (%) was calculated by comparing the mycelium growth on the control and the compound-supplemented plates. The inhibition percentage was calculated as I% = (C − T)/C × 100, where C = mycelium growth in the control medium and T = mycelium growth in the medium added with the tested compound.

## Data Availability

Not applicable.

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
