# Peer review of "Exploration of Novel Scaffolds Targeting Cytochrome b of Pyricularia oryzae"

_ijms, 2023, doi:10.3390/ijms24032705_

Round 1

Reviewer 1 Report

In the present manuscript, Cecilia et.al. tried to identify novel scaffolds with a target of achieving better or comparable inhibition efficiency than the extensively used fungicide Azoxystrobin. They were able to identify one compound that showed a good mycelium inhibition profile for WT and RES P. oryzae. Although the compounds did not show a better profile than Azoxystrobin, the following suggested works may add some improvements over the current work.

Major revisions:

Ø  More details should be included on QoI fungicides and their mechanism of action in the introduction.

Ø  The readers will be benefited to know the importance of the XPG score/MW parameter to identify the promising molecules. Please add an explanation for choosing this parameter in writing (Section 2.1)

Ø  Although it is mentioned by authors in line 100 that 1a is forming π-π interaction with Tyr132 like Metyltetrapole, no docking study has been shown with Metyltetrapole to verify this interaction. Please clarify or mention reference as a justification for this.

Ø  In figure 2, only the structures of the synthesized compounds have been shown. Please include detailed schemes, reagents and % of yield for each step.

Ø  The π-stacking interaction between Tyr280 and compound 3 can’t be seen in figure 1, although mentioned in line 123. Please check.

Ø  Although there are few analogues shown for series 1,2, 5 and 6, no analogues were tested for compound 3. It may be interesting to see the effect of p-Cl/F (w.r.t N-benzyl substitution) and p-Cl/F (w.r.t Hydrazone) in 3 to compare the inhibitory activity with the 4-substitution of other series.

Ø  The rationale behind choosing such a wide range of chain length is not clear, specifically from n=2 to n=4 & n=8. To be more specific, 3, 5a, 6a-6c showed higher inhibitory activity against WT and RES strains (Fig 5). How do the authors explain the correlation between the chain-length and inhibitory activity? The authors may perform some docking studies to check any correlation between them.

Ø  Authors may prepare two bar graphs for WT and RES strains separately for enzymatic assay for better clarification.

Ø  Please clarify the comment made in the discussion section (lines 119-231).

Ø  For the Materials and Methods section:

1.       The HRMS and HPLC of all the newly synthesized compounds should be provided for the purity of the compounds.

2.       The copy of solved 1H and 13C NMR of newly synthesized compounds should be provided in supplementary with proper peak labelling and integration.

3.       The E/Z mix should be separable by column. If it is so, it would be interesting to see the inhibitory activity of E & Z separately.

Ø  The data corresponding to the effect of DMSO on fungal growth (control exp) is important to show to understand the effect of the vehicle.

Ø  It would be helpful to know, why did the authors only use Azoxystrobin for the assays, not the Metyltetrapole along with it?

Minor revisions:

Ø  Please provide the references for the inhibitors mentioned in lines 45-48 in the introduction part.

Ø  Section-4.3.6, line 145- it should be compound 9, not 7. Likewise, in line 455, compound 8 will be 10. Also correct the compound numbers in lines- 452, 456 and 464.

Author Response

In the present manuscript, Cecilia et.al. tried to identify novel scaffolds with a target of achieving better or comparable inhibition efficiency than the extensively used fungicide Azoxystrobin. They were able to identify one compound that showed a good mycelium inhibition profile for WT and RES P. oryzae. Although the compounds did not show a better profile than Azoxystrobin, the following suggested works may add some improvements over the current work.

Major revisions:

-More details should be included on QoI fungicides and their mechanism of action in the introduction.

We added a paragraph in the Introduction on QoI fungicides (lines 56-63).

Strobilurins were introduced to the market in 1996 and still represent one of the most used fungicide classes [11–13]. Their mechanism of action has been studied in great detail; they act within the inner mitochondrial membrane, in particular on complex III by binding to the quinol oxidation site (Qo) of cytochrome b. This binding blocks the electron transfer between cytochrome b and cytochrome c1, which, in turn, leads to an energy deficiency in fungal cells that halts oxidation of the reduced nicotinamide adenine dinucleotide (NADH) and synthesis of adenosine triphosphate (ATP), ultimately leading to cell death [14].

-The readers will be benefited to know the importance of the XPG score/MW parameter to identify the promising molecules. Please add an explanation for choosing this parameter in writing (Section 2.1)

Thanks to the Reviewer’s suggestion, we added a paragraph explaining the choice of this parameter (Materials and Methods, section 4.1).

-Although it is mentioned by authors in line 100 that 1a is forming π-π interaction with Tyr132 like Metyltetrapole, no docking study has been shown with Metyltetrapole to verify this interaction. Please clarify or mention reference as a justification for this.

We added the missing reference (Kunova et al. 2021), where such interactions were characterized in silico.

-In figure 2, only the structures of the synthesized compounds have been shown. Please include detailed schemes, reagents and % of yield for each step.

We replaced the Figure 2 by Scheme 2 with the detailed information requested.

-The π-stacking interaction between Tyr280 and compound 3 can’t be seen in figure 1, although mentioned in line 123. Please check.

We thank the reviewer for pointing out this inconsistency; we updated the figure as images depicting compounds 2a and 3 were erroneously inverted.

-Although there are few analogues shown for series 1,2, 5 and 6, no analogues were tested for compound 3. It may be interesting to see the effect of p-Cl/F (w.r.t N-benzyl substitution) and p-Cl/F (w.r.t Hydrazone) in 3 to compare the inhibitory activity with the 4-substitution of other series.

We thank the reviewer for the comment. We agree that the compound 3 chemotype deserves further and deeper investigation, considering the inhibitory activity on the cytbc1 function in both WT and RES strains. For this reason we planned to prepare a series of analogues to have an insight not only on the substitution pattern of the aromatic rings but also on the role of the hydrazone moiety. This study will be carried out in the near future.

-The rationale behind choosing such a wide range of chain length is not clear, specifically from n=2 to n=4 & n=8. To be more specific, 3, 5a, 6a-6c showed higher inhibitory activity against WT and RES strains (Fig 5). How do the authors explain the correlation between the chain-length and inhibitory activity? The authors may perform some docking studies to check any correlation between them.

We thank the reviewer for the comment. Starting from compounds selected by virtual screening we planned to prepare a series of analogues with a wide range of chain lengths with the aim to expand the SAR study, gathering information about the role of the aliphatic chain linking the two functionalised aromatic rings. Actually, obtained in vitro results showed that chain homologation did not play a key role for the activity.

-Authors may prepare two bar graphs for WT and RES strains separately for enzymatic assay for better clarification.

We thank the reviewer for the precious suggestion. The bar graph was added as suggested.

-Please clarify the comment made in the discussion section (lines 119-231).

We thank the reviewer for rising the point. Actually, the sentence could be misleading and we have removed it for sake of clarity.

For the Materials and Methods section:

-The HRMS and HPLC of all the newly synthesized compounds should be provided for the purity of the compounds.

We have now added a sentence clarifying that all final products showed a purity of >95% by means of NMR spectroscopy.

-The copy of solved 1H and 13C NMR of newly synthesized compounds should be provided in supplementary with proper peak labelling and integration.

All solved 1H NMR and 13C NMR of newly synthesized compounds have been added in supplementary material with proper peak labelling and integration.

The E/Z mix should be separable by column. If it is so, it would be interesting to see the inhibitory activity of E & Z separately.

Unfortunately attempts to separate the isomers were unsuccessful. We have now described more properly NMR spectra in the experimental section.

The data corresponding to the effect of DMSO on fungal growth (control exp) is important to show to understand the effect of the vehicle.

In paragraph 2.4, we have modified the sentences as follows: “It had minimal inhibitory activity on both WT and RES strains (< 10% inhibition of mycelial growth, Supplementary Figure S1). Thus, inhibitory activity of DMSO 1% was subtracted from the inhibition calculation of the tested compounds (see Materials and Methods for details)." and "In this case, the addition of 1% DMSO strongly inhibited the growth of M. pyriformis (ca. 40%) and B. cinerea (ca. 20%), and F. graminearum (< 15%), while it had only a mini-mum inhibitory effect on other fungi (<10%; data not shown). Suprisingly, DMSO seemed to have a positive effect on the growth of F. culmorum (Supplementary Figure S1).”. Moreover, we added a graph showing the DMSO effect on fungal growth in supplementary material.

It would be helpful to know, why did the authors only use Azoxystrobin for the assays, not the Metyltetrapole along with it?

We selected azoxystrobin as a golden standard for our in vitro experiments as this compound is currently used in P. oryzae management. Metyltetraprole, a recently discovered compound (Iwahashi et al. BMC 2020), has been considered only for the development of our 3D model.

Minor revisions:

-Please provide the references for the inhibitors mentioned in lines 45-48 in the introduction part.

The reference for the Fitogest® browser was added, where the information about products registered for disease management can be found (https://fitogest.imagelinenetwork.com/it/). Moreover, individual web pages of the most important fungicide producers were searched for the information, eg. Syngenta (https://www.syngenta.it/riso), BASF (https://www.agro.basf.it/it/Colture/Riso/).

-Section-4.3.6, line 145- it should be compound 9, not 7. Likewise, in line 455, compound 8 will be 10. Also correct the compound numbers in lines- 452, 456 and 464.

We thank the reviewer. All corrections have been made.

Reviewer 2 Report

Whereas the work seems to be carefully done, some few points need attention before publication. 

1) A deeper discussion about theoretical (binding energies, molecular interactions from docking calculations, ...) and experimental data (Enzymatic inhibition of cytochrome bc1 complex by the compounds) could be reported by authors.

2) In addition, the assessment of in vitro absorption, distribution, metabolism, excretion and toxicity (ADME/TOX) parameters of potential inhibitors could be reported in this paper at theoretical or experimental level.

Author Response

Whereas the work seems to be carefully done, some few points need attention before publication. 

-A deeper discussion about theoretical (binding energies, molecular interactions from docking calculations, ...) and experimental data (Enzymatic inhibition of cytochrome bc1 complex by the compounds) could be reported by authors.

Thanks to the Reviewer’s suggestion, we added a paragraph discussing the relevance of binding energies and molecular interactions calculated by virtual screening procedures and we have modified the part regarding the enzymatic inhibition (3. Discussion).

-In addition, the assessment of in vitro absorption, distribution, metabolism, excretion and toxicity (ADME/TOX) parameters of potential inhibitors could be reported in this paper at theoretical or experimental level.

Thanks to the Reviewer’s suggestion, we improved the manuscript by adding Results section 2.5, reporting in silico calculation of some valuable ADME/TOX predictors.

Round 2

Reviewer 1 Report

The authors tried to address all the comments and they added the necessary figures, schemes as asked. There are few points to mention-

Ø  In the supplementary, the authors should provide the name of the NMR solvents in each NMR. Also, the authors should maintain a specific range for showing the NMR, e.g, 0-10 ppm for 1H NMR and 0-200 ppm for 13C NMR ideally.

Ø  The baselines for all the 13C NMR should be corrected.

Ø  Main manuscript, line 256, compound 1a should be 1b.

Ø  Please show the yield of individual E & Z isomers based on the NMR profile obtained.

Ø  The authors should modify the image quality of figure 1.

Author Response

We thank you very much for the reviewer's comments. We have corrected all the points as requested.